# Trends in Urinary and Blood Cadmium Levels in U.S. Adults with or without Comorbidities, 1999–2018

**DOI:** 10.3390/nu14040802

**Published:** 2022-02-14

**Authors:** Jingli Yang, Kenneth Lo, Aimin Yang

**Affiliations:** 1College of Earth and Environmental Sciences, Lanzhou University, Lanzhou 730000, China; yangjl18@lzu.edu.cn; 2Department of Applied Biology and Chemical Technology, The Hong Kong Polytechnic University, 11 Yuk Choi Road, Hung Hom, Kowloon, Hong Kong SAR, China; 3Research Institute for Smart Ageing, The Hong Kong Polytechnic University, 11 Yuk Choi Road, Hung Hom, Kowloon, Hong Kong SAR, China; 4Department of Medicine and Therapeutics, Prince of Wales Hospital, The Chinese University of Hong Kong, Hong Kong SAR, China

**Keywords:** temporal trends, cadmium, comorbidities, cross-sectional study, surveillance

## Abstract

Although cadmium (Cd) exposure has been declining in the United States (U.S.) over the years, the level of exposure for people with pre-existing comorbidities is unclear. This study characterized the trends of blood Cd levels (bCd) (*n* = 44,498) and urinary Cd levels (uCd) (*n* = 15,107) by pre-existing comorbidities among adults participating in the U.S. National Health and Nutrition Examination Survey. We calculated age- and sex-standardized annual geometric mean (GM) levels, and used aJoinpoint regression model to examine the trends over time. The GM levels of bCd declined from 1999–2000 to 2017–2018 survey cycles (from 0.49 to 0.33 μg/L), while women and current smokers had higher levels. Participants with comorbidities had higher bCd and declined over time: cardiovascular disease (CVD) (0.50 to 0.42 μg/L), hypertension (0.49 to 0.35 μg/L), chronic kidney disease (CKD) (0.54 to 0.37 μg), and cancer (0.57 to 0.38 μg) versus those without these comorbidities. We observed the similar pattern of changes for uCd and participants with CVD, CKD, and cancer had higher levels. To conclude, the trend in urinary and blood Cd levels in U.S adults decreased in the past 20 years, and the levels varied by sex, smoking status, and comorbidities.

## 1. Introduction

Cadmium (Cd), a toxic and carcinogenic heavy metal, is mainly emitted to soil, water, and air by non-ferrous metal mining and refining, manufacturing, the application of phosphate fertilizers, fossil fuel combustion, as well as waste incineration and disposal [1]. People can be exposed to Cd from air, diet, drinking water, or by skin contact. Cd mainly accumulates in the kidneys and liver, with estimated half-lives of 6–38 years in kidney and 4–19 years in liver [2]. Urinary Cd levels (uCd, half-life of 15–30 years) mainly reflect cumulative Cd exposure in long-term exposures, whereas blood Cd levels (bCd, half-life of 3–4 months) reflect recent exposures [3]. Since the early 1950s, the hazards of occupational exposure to Cd have been recognized, concerning the toxic effects of Cd exposure in humans and laboratory animals. The International Agency for Research on Cancer (IARC) and the United States (U.S.) Environmental Protection Agency (EPA) have classified Cd as a human carcinogen and a probable human carcinogen, respectively [4].

Mayo Clinic laboratories have suggested that the reference value for Cd in the blood is <5 µg/L for all ages [5], and Cd in urine is <0.6 μg/g creatinine for people aged 18 years and above [6]. Chronic exposure to Cd causes accumulated renal damage. The prevalence of abnormal β2-microglobulin levels increased when Cd levels were higher than reference values [2]. Furthermore, previous studies have found that Cd induces the expression of various pro-atherogenic adhesion molecules on the surface of endothelial cells, thereby facilitating adhesion and migration of immune cells into the vessel wall [7]. Elevated uCd levels were associated with 11% higher risk of stroke (median 0.44 μg/g creatinine versus median 0.10 μg/g creatinine) [1], and 49% higher risk of hypertension (≥5.22 μg/g creatinine versus <1.15 μg/g creatinine) [8]. Moreover, a meta-analysis showed the highest level of urinary Cd increased the risk of mortality from all-cause and cardiovascular disease (CVD) by 44% and 57%, respectively [9]. The human body can detoxify most Cd, but excessive Cd can overload the ability of the liver and kidneys. While many studies have focused on the adverse health effects of Cd, the temporal analysis of Cd exposure in the general population is important for identifying at-risk populations for intervention. Additionally, diet is the primary source of Cd exposure for the general population. According to the U.S. National Health and Nutrition Examination Survey (NHANES) 2007–2012, the foods that contributed most to total Cd intake included lettuce, spaghetti, bread, and potatoes. Deficiency of essential nutrients, such as zinc, manganese, and copper, has been linked with exacerbation of the effects of Cd [10,11].

Ann et al. found that bCd decreased in the Korean general population from 2008 to 2017 [3]. Moreover, there was a declining trend for uCd and bCd in U.S. adults from 1988 to 2018, and higher Cd exposure was associated with women, the elderly, people with poverty, and those with lower education attainment [12]. Although some studies have examined how Cd exposure varied by age, sex, and socioeconomic status, the temporal trend of Cd exposure by pre-existing comorbidities is unclear. Given the close relationship between elevated Cd levels and cardiovascular and kidney diseases, identifying the prevalence of the at-risk population with excessive Cd exposure is important, which has been characterized by the present study among adults who participated in the U.S. NHANES.

## 2. Materials and Methods

### 2.1. Data Source and Study Population

NHANES is a nationally representative, cross-sectional survey of the resident civilian noninstitutionalized U.S. population designed to monitor the health and nutritional status of the entire nation. In this study, we used the continuous NHANES data from 1999 to 2018 among adults. The formulation and review of the NHANES program complies with the U.S. Department of Health and Human services’ policy to protect human research subjects (45 CFR 46, available from https://www.hhs.gov/ohrp/regulations-and-policy/regulations/45-cfr-46/index.html, accessed on 10 June 2021). The National Center for Health Statistics research Ethics Review Board (NCHS ERB) reviewed and approved the study (NCHS ERB protocol number, available from https://www.cdc.gov/nchs/nhanes/irba98.htm, accessed on 10 June 2021). Informed consent was obtained from participants upon recruitment.

### 2.2. Blood and Urinary Cadmium Measurements

Urine and blood samples were processed, stored, and shipped to the Division of Laboratory Sciences, National Center for Environmental Health. Biological samples were stored under appropriate frozen conditions (−20 °C from 1999 to 2006 and −30 °C from 2007 to 2018) until they were shipped to National Center for Environmental Health for test and the Centers for Disease Control and Prevention, Atlanta, GA, for analysis. The bCd were determined on a PerkinElmer Model SIMAA 6000 simultaneous multi-element atomic absorption spectrometer with Zeeman background correction in 1999–2000 and 2001–2002. The bCd concentrations were determined using inductively coupled plasma mass spectrometry (ICP-MS) for blood samples from 2003 to 2018 and for urine samples from 1999 to 2016. This multi-element analytical technique is based on quadrupole ICP-MS technology. The NHANES quality assurance and quality control (QA/QC) protocols meet the 1988 Clinical Laboratory Improvement Act mandates. Detailed instructions on specimen collection and processing, and QA/QC are discussed in the NHANES website (https://www.cdc.gov/nchs/nhanes/index.htm, accessed on 23 June 2021).

### 2.3. Sociodemographic Characteristics

Questionnaires were administered by trained interviewers in the participant’s home to collect demographic information and medical history. Age was categorized as three groups (20–39, 40–59, or ≥60 years old). For adults 20 and older, smoking habits were asked by trained interviewers using a Computer-Assisted Personal Interviewing (CAPI) system in the home. In this study, smoking status was categorized as never smoker (<100 cigarettes in the entire life), former smoker (≥100 cigarettes and not smoking currently) or current smoker (≥100 cigarettes and smoking currently) [13].

### 2.4. Definition for Pre-Existing Comorbidities

Diabetes was defined as the presence of at least one of following: (1) fasting plasma glucose ≥ 7.0 mmol/L (126 mg/dL); (2) hemoglobin A1c ≥ 6.5% (48 mmol/mol); (3) oral glucose tolerance test ≥ 11.1 mmol/L (200 mg/dL); (4) for a patient with classic symptoms of hyperglycemia or hyperglycemic crisis, having a random plasma glucose ≥ 11.1 mmol/L (200 mg/dL) [14]. CKD was defined as GFR < 60 mL/min per 1.73 m^2^ or a urinary albumin-to-creatinine ratio > 30 mg/g [15]. Hypertension was defined as the presence of at least one of the following conditions: (1) systolic blood pressure ≥140 mmHg or diastolic blood pressure ≥ 90 mmHg; (2) current use of medication to treat hypertension; and/or (3) self-reported hypertension [16]. Any CVD was considered to be present at baseline if the participant self-reported prior coronary heart disease (CHD), heart failure (HF), or stroke as informed by a doctor [17]. Cancer was defined as self-reported history of cancer or malignancy.

### 2.5. Statistical Analyses

We calculated age- and sex-standardized annual geometric mean (GM) Cd levels for each calendar year using the total population from 1999 to 2018 as referent. The uCd was divided by urinary creatinine to control the concentration dilution in urine. The GM levels of bCd and uCd were calculated by sex (dichotomized as men or women), smoking status (categorized as current smoker, former smoker, never smoker), and each pre-existing comorbidity (dichotomized as yes or no). We also estimated the prevalence of bCd ≥ 5 µg/L (reference values) [5] and uCd ≥ 0.6 μg/g creatinine [6], and the χ^2^ test was used to compare differences between groups. When the bCd and uCd were below the detection limit, the values were the limit of detection divided by the square root of 2.

We used a Joinpoint regression model to examine trends in bCd and uCd over time. This software uses permutation tests to identify points where linear trends change significantly in either direction or magnitude. It calculated the average annual percentage change (AAPC) and 95% confidence interval (95% CI) for the full study period and the annual percentage change for each linear trend segment detected. Tests of coincidence were performed in pairwise comparison to see whether the changing trend of bCd and uCd was different across the different subgroups. Two-sided *p* values < 0.05 were considered statistically significant. All statistical analyses were completed using R version 4.0.3 software (R Foundation for Statistical Computing, Vienna, Austria) and the NCI Joinpoint Regression Program version 4.8.0.1 (National Cancer Institute: Rockville, MD, USA, 2020).

## 3. Results

A total of 101,316 participants took part in the NHANES survey from 1999 to 2018. For analyzing the trend in bCd, we excluded participants who were missing bCd data (*n* = 24,927) and aged < 20 years (*n* = 31,891). Finally, 44,498 participants were included in the bCd analysis (Appendix A). During these 10 survey cycles, the prevalence of men varied between 46.7% and 49.3%. The prevalence of diabetes varied between 11.8% and 21.8%, hypertension varied between 38.7% and 46.9%, CKD varied between 10.7% and 13.8%, cancer varied between 1.3% and 9.8%, and any CVD varied between 9.4% and 13.3% (Table 1). For analyzing the trend in uCd, we excluded participants who did not have uCd data (*n* = 77,546), or those aged < 20 years (*n* = 8663). Finally, 15,107 participants were included in uCd analysis (Appendix A). The distribution prevalence of other subgroups was similar to that of bCd analysis.

As shown in Table 2, the prevalence of bCd higher than reference value (5 μg/L) was 0.2% among overall participants. There were 2595 (17.2%) participants whose urinary Cd levels were greater than the reference value (≥0.6 μg/g creatinine). The prevalence of elevated uCd in participants with cancer (23.2% versus 15.4%), CKD (26.7% versus 15.8%), hypertension (23.5% versus 12.7%), and any CVD (32.3% versus 15.4%) was significantly higher than participants without these comorbidities (*p* value < 0.001). More than 30% of participants had at least one CVD, including CHD (33.4%), stroke (33.1%), and HF (32.7%) and had uCd over the reference values. There were 30.5% of current smokers whose urinary Cd levels were greater than the reference values.

The bCd levels declined among overall participants from 1999–2000 to 2017–2018 (Figure 1). The standardized GM of bCd levels decreased from 0.49 μg/L in 1999–2000 to 0.33 μg/L in 2017–2018 (AAPC = −2.0, 95%CI: −2.5 to −1.4) among overall participants (Appendix A, Figure 1). The bCd levels were higher among women than men during these 10 survey cycles (Figure 1A, Table 3). The bCd levels were the highest among current smokers and did not change significantly during these 10 survey cycles (Figure 1B, Table 3). Compared with participants without these comorbidities, participants with pre-existing comorbidities had higher bCd levels and declined over time. The standardized GM levels of bCd in 1999–2000 versus 2017–2018 were 0.50 μg/L versus 0.42 μg/L for any CVD (AAPC = −1.3, 95%CI: −2.2 to −0.4), 0.49 μg/L versus 0.35 μg/L for hypertension (AAPC = −1.5, 95%CI: −1.9 to −1.0), 0.54 μg/L versus 0.37 μg/L for CKD (AAPC = 2.0, 95%CI: −2.7 to −1.3), and 0.57 μg/L versus 0.38 μg/L for cancer (AAPC = −1.4, 95%CI: −2.1 to −0.8) (Figure 2, Table 3). The bCd levels were higher among participants with CHD and HF than among their counterparts (Appendix A, Table 3). The values of bCd levels are summarized in Appendix A.

Similar to bCd, the uCd levels declined among overall participants from 1999–2000 to 2015–2016, and women had higher uCd levels (Figure 3A, Table 3). The standardized GM levels of uCd in 1999–2000 versus 2015–2016 were 0.61 μg/g creatinine versus 0.40 μg/g creatinine among current smokers (AAPC = −2.4, 95%CI: −3.2 to −1.6), but the trend was declining during all of the survey cycles (Figure 3B, Table 3). Compared with participants without these comorbidities, participants with pre-existing comorbidities had higher uCd levels, which declined over time. The standardized GM levels of uCd in 1999–2000 versus 2015–2016 were 0.44 μg/g creatinine versus 0.30 μg/g creatinine for any CVD (AAPC = −1.7, 95%CI: −3.5 to 0.1), 0.38 μg/g creatinine versus 0.26 μg/g creatinine for CKD (AAPC = −2.3, 95%CI: −3.5 to −1.2), and 0.42 μg/g creatinine versus 0.24 μg/g creatinine for cancer (AAPC = −2.9, 95%CI: −4.7 to −1.0) (Figure 4, Table 3). The participants with any CVD, such as stroke, CHD, and HF, had higher uCd levels than participants without these comorbidities (Appendix A, Table 3). The uCd levels are summarized in Appendix A.

## 4. Discussion

The levels of Cd in blood and urine declined from 1999–2000 to 2017–2018 survey cycles. The levels of bCd and uCd were higher among women than men, and higher among current smokers than former smokers and never smokers. The levels of bCd and uCd among participants with cancer, CKD, hypertension, and any CVD were higher than participants without these diseases, and the levels declined during these 10 survey cycles. The prevalence of uCd over reference values in participants with cancer, CKD, and any CVD, especially in current smokers, was significantly higher than their counterparts. Moreover, the uCd and bCd levels among former smokers were significantly lower than current smokers.

The Cd levels (GM < 0.49 μg/L) in blood from 1999 to 2018 were lower than the reference values (<5.0 μg/L) [5] in overall participants. When comparing with studies conducted in Italy (GM = 0.53 μg/L) [18], Germany (Mean = 0.57 μg/L) [19], Canada (Reference values = 0.83 μg/L) [20], and Australia ((Mean = 0.8 μg/L) [21], we observed a lower bCd level among U.S. adults in the present study. The Cd levels (GM < 0.34 μg/g creatinine) in urine from 1999 to 2016 were lower than the reference values (<0.6 μg/g creatinine) [6] among people aged > 20 years overall. When comparing with studies conducted in other countries, the GM of uCd in urine was 0.61 μg/g creatinine in Korea [22], 0.37 μg/L in Northern France [23], and 1.3 μg/L in Canada [20], which are higher than the uCd levels among U.S. adults in the current study, implying that Cd levels in the U.S. adults were at a lower level.

Despite lower bCd and uCd levels among U.S. adults than among those in other countries, the levels of bCd and uCd were higher among participants with cancer, CKD, hypertension, and any CVD. In other words, people with pre-existing chronic diseases are more vulnerable to high levels of Cd exposure, and they were at high risk of exposure to Cd and its potential adverse health effects. Chronic exposure to lower levels of Cd can accumulate in kidneys, and may induce kidney disease, and adverse hepatic and bone health [24]. Cd exposure can relate to early signs of renal damage, proteinuria, calcium loss, and tubular lesion [25]. Since Cd toxicity is dependent on its concentration in the kidneys, its adverse effects are typically not observed in shorter durations. In addition, the health effects of Cd exposure have been reported to be an increased risk of any CVD [26], stroke [27], and hypertension [28]. Results from several experimental studies support the effects of Cd toxicity on cardiovascular disease risk, including endothelial cell death, oxidative stress, smooth muscle cell accumulation, and vascular inflammation [29]. Cd toxicity is interrelated with various comorbidities and may elevate the risk of mortality in the long-term. Duan et al. showed that blood Cd increased the risk of mortality from all-cause, CVD, and cancer by 32%, 27%, and 49%, respectively, in U.S. adults [30]. Cd can change DNA methylation [31], inhibit DNA repair [32], and increase cell apoptosis [33], which may be the cause of cancer. However, considering the complexity of the causative factors of these comorbidities, the physiological mechanisms behind them warrants further investigation.

Our analysis found that Cd levels were higher in women than in men, which was consistent with the results found in a Spanish study for uCd [34] and Italy for bCd [18]. In addition, a meta-analysis also found uCd levels were higher in women than in men in all studies that reported results stratified by sex [9]. The possible reason for the higher body burden of Cd in women is related to higher gastrointestinal absorption of Cd at low iron stores [35]. The absorption of Cd in the human body is closely related to iron transporters in small intestinal epithelial cells, such as divalent metal transporter (DMT-1) and high iron transporter (FPN1). When the body is deficient in iron, the surface of iron transfer is upregulated, leading to increased absorption and accumulation of Cd [36].

Moreover, we found that uCd and bCd levels among current smokers were significantly higher than never smokers, but the uCd and bCd levels among former smokers were lower than current smokers. Studies in U.S. population showed the rate of current smokers has declined from 20.9% (approximately 21 of every 100 adults) in 2005 to 14.0% (14 of every 100 adults) in 2019, and the prevalence of ever smokers who have quit has increased [37]. This is consistent with the decrease in uCd found in this study from 1999 to 2018. A study also found that the rate of current smokers was the highest among people of non-Hispanic races, with a general education development certificate, or with a low annual household income [37]. It has been estimated that tobacco smokers are exposed to 1.7 μg Cd per cigarette, and about 10% is inhaled when smoked [2]. Non-smokers may also be exposed to Cd in cigarettes via second-hand smoke [2]. In our analysis, former smokers had lower Cd levels than current smokers, which indicates the importance of smoking cessation, even for people who have smoked before.

As demonstrated by our analysis, there are several at-risk populations in which exposure to Cd need to be reduced. The Agency for Toxic Substances and Disease Registry [2] has put forward several suggestions—namely, (1) do not smoke tobacco products; (2) adopt good occupational hygiene, such as bathing and changing clothes before returning home; (3) avoid Cd contaminated areas and food, such as hazardous waste sites; (4) dispose of cadmium containing products properly, such as recycling old batteries whenever possible. In addition, patients with Cd poisoning can be treated with gastrointestinal irrigation, supportive therapy, and chemical decontamination with conventional chelation therapy [25].

There were some limitations in this study. First, biological samples were only collected at single time point and may have introduced bias. Second, we were not able to establish a causal relationship between Cd levels and disease due to the repeated cross-sectional design. Third, the history of comorbidities was self-reported, and the prevalence of comorbidities may be underestimated in this study. Our study also has some strengths. The NHANES is a nationally representative survey and provides nationally representative estimates. We estimated trends over the past over 20 years in both blood and urine Cd levels by pre-existing comorbidities. We used uCd levels as divided by urinary creatinine to control the concentration dilution of urine.

## 5. Conclusions

During the past 20 years from 1999 to 2018, the overall trends of Cd levels in blood and urine have declined among U.S. adults but varied by sex, smoking status, and pre-existing comorbidities. Considering that environmental exposure to heavy metals such as Cd plays an important role in the development of chronic diseases, further studies are needed to evaluate the associations between heavy metals and the risk of chronic diseases.

## Figures and Tables

**Figure 1 nutrients-14-00802-f001:**
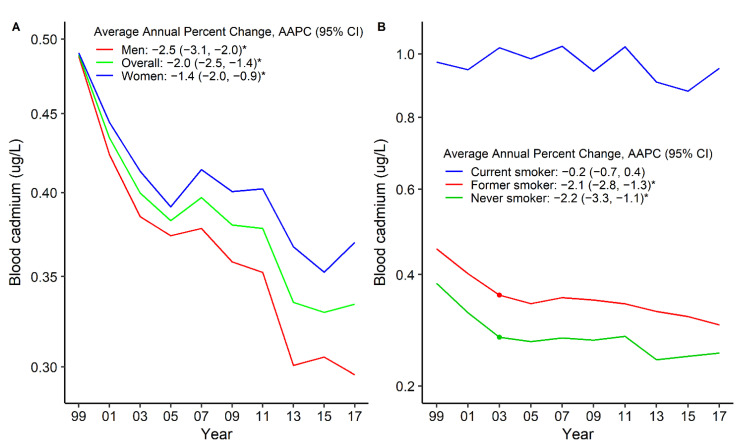
Trends in standardized geometric mean of blood cadmium levels by sex (**A**) and smoking status (**B**). * The solid line indicated the age– and sex–standardized geometric mean of blood cadmium levels. Points indicate the change points (joinpoints) in trends detected by the Joinpoint regression model. The APPC is significantly different from zero at the α = 0.05 level. Data are from the U.S. National Health and Nutrition Examination Survey 1999–2018.

**Figure 2 nutrients-14-00802-f002:**
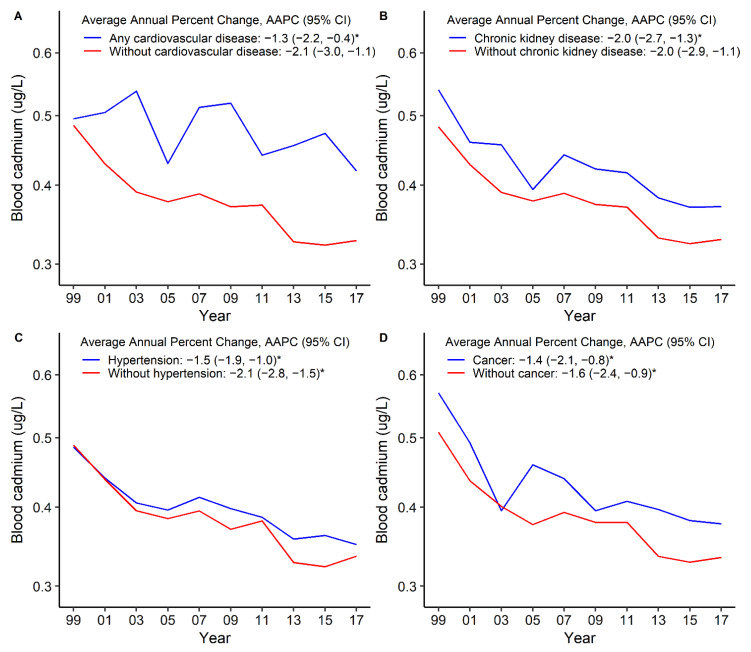
Trends in standardized geometric mean of blood cadmium levels by any cardiovascular disease (**A**), chronic kidney disease (**B**), hypertension (**C**), and cancer (**D**). * The solid line indicated the age– and sex–standardized geometric mean of blood cadmium levels. Points indicate the change points (joinpoints) in trends detected by the Joinpoint regression model. The APPC is significantly different from zero at the α = 0.05 level. Data are from the U.S. National Health and Nutrition Examination Survey 1999–2018.

**Figure 3 nutrients-14-00802-f003:**
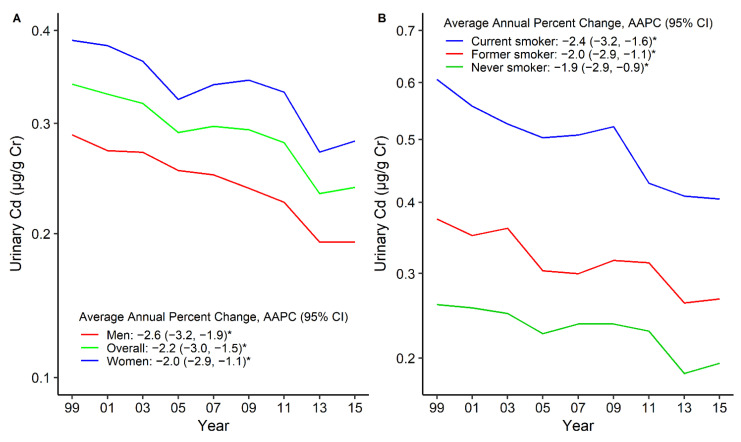
Trends in standardized geometric mean of urinary cadmium levels by sex (**A**) and smoking status (**B**). * The solid line indicated the age– and sex–standardized geometric mean of urinary cadmium levels. Points indicate the change points (joinpoints) in trends detected by the Joinpoint regression model. The APPC is significantly different from zero at the α = 0.05 level. Data are from the U.S. National Health and Nutrition Examination Survey 1999–2016.

**Figure 4 nutrients-14-00802-f004:**
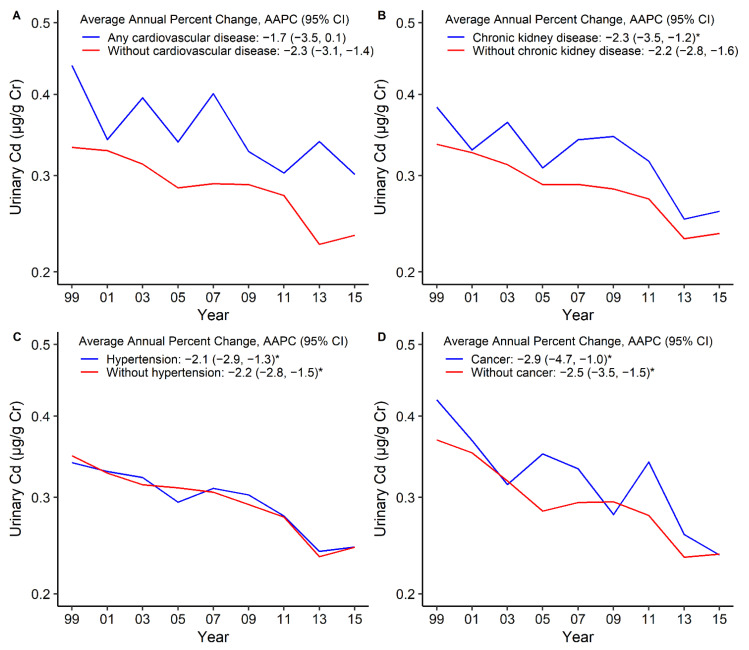
Trends in standardized geometric mean of urinary cadmium levels by any cardiovascular disease (**A**), chronic kidney disease (**B**), hypertension (**C**), and cancer (**D**). * The solid line indicated the age– and sex–standardized geometric mean of urinary cadmium levels. Points indicate the change points (joinpoints) in trends detected by the Joinpoint regression model. The APPC is significantly different from zero at the α = 0.05 level. Data are from the U.S. National Health and Nutrition Examination Survey 1999–2016.

**Table 1 nutrients-14-00802-t001:** Characteristics of participants in the U.S. NHANES 1999–2018.

	1999–2000	2001–2002	2003–2004	2005–2006	2007–2008	2009–2010	2011–2012	2013–2014	2015–2016	2017–2018	Overall
Blood cadmium analysis											
N	4207	4772	4525	4509	5364	5765	5030	2695	2610	5021	44,498
Men, *n* (%)	1966 (46.7)	2259 (47.3)	2182 (48.2)	2158 (47.9)	2622 (48.9)	2784 (48.3)	2480 (49.3)	1290 (47.9)	1277 (48.9)	2417 (48.1)	21,435 (48.2)
Smoking status, *n* (%)											
Never smoker	2234 (53.1)	2445 (51.2)	2267 (50.1)	2374 (52.7)	2816 (52.5)	3085 (53.5)	2872 (57.1)	1533 (56.9)	1522 (58.3)	2905 (57.9)	24,053 (54.1)
Former smoker	1116 (26.5)	1262 (26.4)	1238 (27.4)	1145 (25.4)	1357 (25.3)	1417 (24.6)	1161 (23.1)	627 (23.3)	608 (23.3)	1210 (24.1)	11,141 (25.0)
Current smoker	857 (20.4)	1065 (22.3)	1020 (22.5)	990 (21.9)	1191 (22.2)	1263 (21.9)	997 (19.8)	535 (19.8)	480 (18.4)	906 (18.0)	9304 (20.9)
Diabetes, *n* (%)	509 (12.1)	562 (11.8)	622 (13.7)	655 (14.5)	1029 (19.2)	1043 (18.1)	951 (18.9)	465 (17.3)	564 (21.6)	1094 (21.8)	7494 (16.8)
Chronic kidney disease, *n* (%)	569 (13.5)	588 (12.3)	527 (11.6)	549 (12.2)	719 (13.4)	614 (10.7)	641 (12.7)	291 (10.8)	331 (12.7)	692 (13.8)	5521 (12.4)
Hypertension, *n* (%)	1729 (41.1)	1879 (39.4)	1945 (43.0)	1747 (38.7)	2314 (43.1)	2359 (40.9)	2102 (41.8)	1144 (42.4)	1127 (43.2)	2357 (46.9)	18,703 (42.0)
Any cardiovascular disease, *n* (%)	444 (10.6)	508 (10.6)	603 (13.3)	498 (11.0)	632 (11.8)	614 (10.7)	514 (10.2)	252 (9.4)	276 (10.6)	619 (12.3)	4960 (11.1)
Coronary heart disease, *n* (%)	315 (7.5)	378 (7.9)	420 (9.3)	337 (7.5)	410 (7.6)	428 (7.4)	313 (6.2)	172 (6.4)	180 (6.9)	418 (8.3)	3371 (7.6)
Stroke, *n* (%)	141 (3.4)	148 (3.1)	186 (4.1)	175 (3.9)	222 (4.1)	206 (3.6)	202 (4.0)	83 (3.1)	102 (3.9)	242 (4.8)	1707 (3.8)
Heart failure, *n* (%)	126 (3.0)	138 (2.9)	166 (3.7)	152 (3.4)	187 (3.5)	158 (2.7)	170 (3.4)	74 (2.7)	80 (3.1)	179 (3.6)	1430 (3.2)
Cancer, n (%)	56 (1.3)	78 (1.6)	176 (3.9)	266 (5.9)	485 (9.0)	567 (9.8)	421 (8.4)	219 (8.1)	225 (8.6)	461 (9.2)	2954 (6.6)
Urinary cadmium analysis											
N	1299	1560	1532	1520	1857	2019	1715	1811	1794	NA	15,107
Men, *n* (%)	620 (47.7)	737 (47.2)	743 (48.5)	736 (48.4)	922 (49.6)	976 (48.3)	867 (50.6)	878 (48.5)	886 (49.4)	NA	7365 (48.8)
Smoking status, *n* (%)											
Never smoker	671 (51.6)	787 (50.5)	792 (51.7)	799 (52.6)	985 (53.1)	1085 (53.7)	986 (57.5)	1019 (56.3)	1030 (57.4)	NA	8154 (54.0)
Former smoker	349 (26.9)	411 (26.3)	420 (27.4)	397 (26.1)	460 (24.8)	503 (24.9)	395 (23.0)	426 (23.5)	427 (23.8)	NA	3788 (25.1)
Current smoker	279 (21.5)	362 (23.2)	320 (20.9)	324 (21.3)	412 (22.2)	431 (21.3)	334 (19.5)	366 (20.2)	337 (18.8)	NA	3165 (20.9)
Diabetes, *n* (%)	159 (12.2)	191 (12.2)	222 (14.5)	217 (14.3)	346 (18.6)	377 (18.7)	327 (19.1)	305 (16.8)	366 (20.4)	NA	2510 (16.6)
Chronic kidney disease, *n* (%)	173 (13.3)	199 (12.8)	187 (12.2)	199 (13.1)	256 (13.8)	230 (11.4)	221 (12.9)	201 (11.1)	224 (12.5)	NA	1890 (12.5)
Hypertension, *n* (%)	525 (40.4)	615 (39.4)	697 (45.5)	560 (36.8)	785 (42.3)	824 (40.8)	709 (41.3)	754 (41.6)	753 (42.0)	NA	6222 (41.2)
Any cardiovascular disease, n (%)	146 (11.2)	163 (10.4)	213 (13.9)	170 (11.2)	204 (11.0)	186 (9.2)	157 (9.2)	170 (9.4)	181 (10.1)	NA	1590 (10.5)
Coronary heart disease, *n* (%)	107 (8.2)	127 (8.1)	144 (9.4)	117 (7.7)	129 (6.9)	124 (6.1)	104 (6.1)	113 (6.2)	120 (6.7)	NA	1085 (7.2)
Stroke, *n* (%)	43 (3.3)	45 (2.9)	67 (4.4)	63 (4.1)	75 (4.0)	65 (3.2)	56 (3.3)	58 (3.2)	69 (3.8)	NA	541 (3.6)
Heart failure, *n* (%)	46 (3.5)	46 (2.9)	56 (3.7)	53 (3.5)	58 (3.1)	49 (2.4)	51 (3.0)	46 (2.5)	49 (2.7)	NA	454 (3.0)
Cancer, *n* (%)	18 (1.4)	22 (1.4)	63 (4.1)	98 (6.4)	162 (8.7)	222 (11.0)	138 (8.0)	146 (8.1)	161 (9.0)	NA	1030 (6.8)

NAHAES, U.S. National Health and Nutrition Examination Survey; NA, not available.

**Table 2 nutrients-14-00802-t002:** The prevalence of blood cadmium levels ≥ 5 μg/L and urinary cadmium levels ≥ 0.6 μg/g creatinine in the U.S. NHANES.

	Blood Cadmium, μg/L			Urinary Cadmium, μg/g Creatinine		
Group, *n* (%)	<5.0	≥5.0	*p* Values	<0.6	≥0.6	*p* Values
Overall	44,429 (99.8)	69 (0.2)	--	12,507 (82.8)	2595 (17.2)	--
Sex			0.307			<0.001
Men	21,406 (99.9)	29 (0.1)		6407 (87.0)	955 (13.0)	
Women	23,023 (99.8)	40 (0.2)		6100 (78.8)	1640 (21.2)	
Smoking status			<0.001			<0.001
Never smoker	24,031 (100)	0 (0.0)		7345 (90.2)	798 (9.8)	
Former smoker	11,129 (100)	0 (0.0)		2952 (78.0)	832 (22.0)	
Current smoker	9229 (99.3)	69 (0.7)		2198 (69.5)	964 (30.5)	
Chronic kidney disease			0.385			<0.001
Yes	5510 (99.8)	11 (0.2)		1385 (73.3)	505 (26.7)	
No	38,641 (99.9)	58 (0.1)		11,122 (84.2)	2090 (15.8)	
Hypertension			0.087			<0.001
Yes	18,681 (99.9)	22 (0.1)		4756 (76.5)	1463 (23.5)	
No	25,735 (99.8)	47 (0.2)		7747 (87.3)	1130 (12.7)	
Diabetes			0.244			<0.001
Yes	7486 (99.9)	8 (0.1)		1927 (84.0)	582 (16.0)	
No	36,943 (99.8)	61 (0.2)		10,580 (76.8)	2013 (23.2)	
Cardiovascular disease			0.099			<0.001
Yes	4948 (99.8)	12 (0.2)		1075 (67.7)	514 (32.3)	
No	39,476 (99.9)	57 (0.1)		11,432 (84.6)	2081 (15.4)	
Coronary heart disease			0.086			<0.001
Yes	3362 (99.7)	9 (0.3)		722 (66.6)	362 (33.4)	
No	41,053 (99.9)	60 (0.1)		11,784 (84.1)	2233 (15.9)	
Stroke			0.684			<0.001
Yes	1705 (99.9)	2 (0.1)		361 (66.9)	179 (33.1)	
No	42,668 (99.8)	67 (0.2)		12,135 (83.4)	2412 (16.6)	
Heart failure			0.010			<0.001
Yes	1424 (99.6)	6 (0.4)		305 (67.3)	148 (32.7)	
No	42,858 (99.9)	63 (0.1)		12,174 (83.3)	2432 (16.7)	
Cancer			0.182			<0.001
Yes	2947 (99.8)	7 (0.2)		791 (76.8)	239 (23.2)	
No	28,906 (99.9)	40 (0.1)		8444 (84.6)	1540 (15.4)	

NAHAES, U.S. National Health and Nutrition Examination Survey. Blood cadmium data come from NHANES 1999–2018, urinary cadmium data come from NHANES 1999–2016.

**Table 3 nutrients-14-00802-t003:** Pairwise comparison of trend in blood and urinary cadmium levels grouped by sociodemographic characteristics and comorbidities.

Cohort 1	Cohort 2	Numerator Degrees of Freedom	Denominator Degrees of Freedom	Number of Permutations	*p*-Value	Coincidence
Blood cadmium analysis						
Men	Women	2	16	4500	0.003	Rejected
Never smoker	Former smoker	4	12	4500	<0.001	Rejected
Never smoker	Current smoker	4	12	4500	<0.001	Rejected
Former smoker	Current smoker	4	12	4500	<0.001	Rejected
Chronic kidney disease	Without chronic kidney disease	4	12	4500	0.002	Rejected
Hypertension	Without hypertension	4	12	4500	<0.001	Rejected
Diabetes	Without diabetes	2	16	4500	0.051	Failed to reject
Any cardiovascular disease	Without cardiovascular disease	4	12	4500	0.003	Rejected
Coronary heart disease	Without coronary heart disease	2	16	4500	0.004	Rejected
Stroke	Without stroke	4	12	4500	0.003	Reject
Heart failure	Without heart failure	2	16	4500	0.054	Failed to reject
Cancer	Without cancer	2	16	4500	0.014	Rejected
Urinary cadmium analysis						
Men	Women	2	14	4500	0.002	Rejected
Never smoker	Former smoker	2	14	4500	<0.001	Rejected
Never smoker	Current smoker	2	14	4500	0.002	Rejected
Former smoker	Current smoker	2	14	4500	0.002	Rejected
Chronic kidney disease	Without chronic kidney disease	2	14	4500	<0.001	Rejected
Hypertension	Without hypertension	4	10	4500	0.736	Failed to reject
Diabetes	Without diabetes	4	10	4500	0.357	Failed to reject
Any cardiovascular disease	Without cardiovascular disease	2	14	4500	<0.001	Rejected
Coronary heart disease	Without coronary heart disease	2	14	4500	<0.001	Rejected
Stroke	Without stroke	2	14	4500	0.009	Rejected
Heart failure	Without heart failure	2	14	4500	0.002	Rejected
Cancer	Without cancer	2	14	4500	0.038	Rejected

## Data Availability

All relevant data are within the manuscript.

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
