# Peer review of "Trends in Urinary and Blood Cadmium Levels in U.S. Adults with or without Comorbidities, 1999–2018"

_nutrients, 2022, doi:10.3390/nu14040802_

Round 1
Reviewer 1 Report
In my opinion, the paper is adequately designed and properly written. My only concern is that I don't feel the paper aims for an adequate journal. I think the International Journal of Environmental Research and Public Health would be much more suitable amongst the journals of the current publisher.
Minor comments
Line 46 Cd accumulates in blood vessels. I think this idea needs a little bit of development here (maybe 1-2 sentences).
Line 79 "samples WERE"
Limitations: Wasn't the use of questionnaires for some morbidities in the initial NHANES survey a limitation?
Author Response
Reviewer 1:
In my opinion, the paper is adequately designed and properly written. My only concern is that I don't feel the paper aims for an adequate journal. I think the International Journal of Environmental Research and Public Health would be much more suitable amongst the journals of the current publisher.
Responses: Thank you for the comments. There is growing evidence on the link between environmental factors including heavy metals such as Cd and risk of chronic diseases. Diet is the primary source of Cd exposure for most individuals. Previous studies showed that some nutrients which are present in diet may modify susceptibility to Cd exposure. Deficientness of essential nutrients has also been linked to exacerbate the effects of Cd. In this study, we did not evaluate the interaction between Cd exposure and nutrients with risk of co-morbidities such as CVD, hypertension, and CKD in this manuscript, however, we observed both blood and urinary Cd levels varied by these co-morbidities, implicating that environmental exposure to heavy metals such as Cd may play important role in the development of those chronic disease. We also mentioned this point in “Introduction” and “Discussion” sections of the revised manuscript.
Additionally, this special issue – “Minerals Metabolism and Human Health” focus on nutritional and non-nutritional factors that may affect mineral status including diet and environmental factors. We believe that the findings of this manuscript will be of interest to the readers of the journal. Thank you.
Minor comments
Line 46 Cd accumulates in blood vessels. I think this idea needs a little bit of development here (maybe 1-2 sentences).
Responses: We have elaborated on this sentence accordingly.
Line 79 "samples WERE"
Responses: We have corrected the typo accordingly.
Limitations: Wasn't the use of questionnaires for some morbidities in the initial NHANES survey a limitation?
Responses: Thank you for your comment. We have added this point into the limitation.
Reviewer 2 Report
Dear Authors
In my opinion the theme of the article is innovate and very interesting and important for the readers of the magazine.
The authors studied the trends of blood Cd levels and urinary Cd levels (uCd) among adults (> 15,000) that participated in U.S. National Health and Nutrition Examination Survey, stratified by their pre-existing co-morbidities.
The levels of Cd in blood and urine were declined from 1999–2000 survey cycle to 2017‑2018.
This research found the levels of urinary and blood Cd were in higher levels among participants with cancer, CKD, hypertension, and any CVD, as well as smokers.
High-risk population with excessive Cd exposure has been identified for controlling their exposure levels.
The manuscript under revision is well structured; the language is correct and clear. The title and abstract clearly describe the content of the manuscript. I suggest the authors to include one or two paragraphs about Cd remediation.
Best regards
Author Response
Reviewer 2:
In my opinion the theme of the article is innovate and very interesting and important for the readers of the magazine.
The authors studied the trends of blood Cd levels and urinary Cd levels (uCd) among adults (> 15,000) that participated in U.S. National Health and Nutrition Examination Survey, stratified by their pre-existing co-morbidities. The levels of Cd in blood and urine were declined from 1999–2000 survey cycle to 2017‑2018. This research found the levels of urinary and blood Cd were in higher levels among participants with cancer, CKD, hypertension, and any CVD, as well as smokers. High-risk population with excessive Cd exposure has been identified for controlling their exposure levels.
The manuscript under revision is well structured; the language is correct and clear. The title and abstract clearly describe the content of the manuscript.
Responses: We sincerely appreciate the positive comments. I suggest the authors to include one or two paragraphs about Cd remediation.
Responses: Thanks for your valuable suggestions. We have presented the Cd remediation strategies as proposed by Agency for Toxic Substances and Disease Registry in the revised manuscript (in the second last paragraph of discussion section).